# Role of T cells during the cerebral infection with *Trypanosoma brucei*

**Gabriela C. Olivera**[1¤a], **Leonie Vetter**[1], **Chiara Tesoriero**[2¤b], **Federico Del Gallo**[2¤c], **Gustav Hedberg**[1], **Juan Basile**[1], **Martin E. Rottenberg**[1] *

**1** Department of Microbiology, Tumor and Cell Biology, Karolinska Institutet, Stockholm, Sweden,
**2** Department of Neuroscience, Biomedicine and Movement Sciences, University of Verona, Verona, Italy

¤a Current address: Department of Molecular Biosciences, Stockholm University, Stockholm, Sweden
¤b Current address: Department of Biotechnology, University of Verona, Verona, Italy
¤c Current address: School of Pharmacy, University of Camerino, Macerata, Italy
* Martin.Rottenberg@ki.se

**Data Availability Statement:** All relevant data are within the manuscript and its Supporting Information files.

## Abstract

The infection by *Trypanosoma brucei brucei* (*T.b.b.*), a protozoan parasite, is characterized by an early-systemic stage followed by a late stage in which parasites invade the brain parenchyma in a T cell-dependent manner.

Here we found that early after infection effector-memory T cells were predominant among brain T cells, whereas, during the encephalitic stage T cells acquired a tissue resident memory phenotype ($T_{RM}$) and expressed PD1. Both CD4 and CD8 T cells were independently redundant for the penetration of *T.b.b.* and other leukocytes into the brain parenchyma. The role of lymphoid cells during the *T.b.b.* infection was studied by comparing T- and B-cell deficient *rag1*[-/-] and WT mice. Early after infection, parasites located in circumventricular organs, brain structures with increased vascular permeability, particularly in the median eminence (ME), paced closed to the sleep-wake regulatory arcuate nucleus of the hypothalamus (Arc). Whereas parasite levels in the ME were higher in *rag1*[-/-] than in WT mice, leukocytes were instead reduced. *Rag1*[-/-] infected mice showed increased levels of *meca32* mRNA coding for a blood /hypothalamus endothelial molecule absent in the blood-brain-barrier (BBB). Both immune and metabolic transcripts were elevated in the ME/Arc of WT and *rag1*[-/-] mice early after infection, except for *ifng* mRNA, which levels were only increased in WT mice. Finally, using a non-invasive sleep-wake cycle assessment method we proposed a putative role of lymphocytes in mediating sleep alterations during the infection with *T.b.b.*

Thus, the majority of T cells in the brain during the early stage of *T.b.b.* infection expressed an effector-memory phenotype while $T_{RM}$ cells developed in the late stage of infection. T cells and parasites invade the ME/Arc altering the metabolic and inflammatory responses during the early stage of infection and modulating sleep disturbances.

## Author summary

*Trypanosoma brucei* (*T.b.*) causes an early systemic and a late encephalitic infection characterized by sleep alterations. In rodent models, brain invasion by *T.b. brucei* (*T.b.b.*) is

**Funding:** The study was supported by a project grant from ERA-NET Neuron, Swedish Research Council 529-2014-7552 and the Karolinska Institutet. The funders had no role in study design, data collection and analysis, decision to publish, or preparation of the manuscript.

**Competing interests:** The authors have declared that no competing interests exist.

strictly dependent on T cells. However, an in-depth characterization of T cell functions and phenotypes in the outcome of *T.b.b.* infection is still lacking.

Here we found that during the early stage of infection of mice, most brain T cells differentiated into memory cells, and acquired a tissue-resident memory phenotype during the encephalitic stage. CD4 and CD8 T cells were redundant for the invasion of other T cells and parasites into the brain. Early after infection *T.b.b.* and leukocytes invade different circumventricular organs (brain areas that lack a blood-brain barrier) including the median eminence (ME) located close to sleep-regulating arcuate nucleus (Arc). *T.b.b.* infection induced the expression of immune and metabolic molecules in this area. Lymphocytes modulated 1) the levels of invading parasites and leukocytes in the ME; 2) the structure of the blood/ hypothalamus interphase and 3) the expression of IFN-γ in the ME/Arc early after infection. Lymphocytes may also be involved in the regulation of sleep alterations observed in African trypanosomiasis.

## Introduction

The extracellular protozoan parasite *Trypanosoma brucei* (*T.b.*) is the causative agent of African trypanosomiasis, an infectious disease that affects both humans and animals. During the early stage of infection, *T.b.* overruns the hemo-lymphatic system. The early stage is followed by a late meningo-encephalitic stage in which severe signs of the central nervous system (CNS) involvement are observed [1,2]. In a mouse model of infection, *T.b. brucei* (*T.b.b.*) and leukocytes cross the blood-brain barrier (BBB) and enter the brain parenchyma [3]. Within the CNS, the activation of leukocytes and resident cells probably contributes to the brain disease [4].

We have previously observed that *T.b.b.* brain invasion depends on the presence of T cells as shown by the absence of parasites in the brain of *rag1*[-/-] mice, lacking mature T and B cells. The parasite penetration was restored after the adoptive transfer of T cells into *rag1*[-/-] mice [3,5]. The secretion of interferon (IFN)-γ by T cells [3], of the IFN-γ-dependent chemokine CXCL10 by astrocytes [6], and of tumor necrosis factor (TNF) by T cells and macrophages [5,7] have been shown to be required for parasite and leukocyte invasion of the brain. An unresolved question is how inflammatory cells are initially recruited into the brain parenchyma during *T.b.b.* infection. Primary brain penetration of parasites or parasite-derived innate immune receptor agonists should theoretically be required for further recruitment of inflammatory cells and penetration through the BBB.

T cells become activated in draining lymph nodes and migrate to different tissues for combating infection. Activated T cells display terminal effector functions or they can further differentiate into memory T cells. The latter might recirculate between peripheral and lymphoid organs and remain in lymphoid organs or express retention markers in the periphery including the brain with recognized protective or detrimental activities. The importance of tissue-resident memory T cells (T$_{RM}$) in the brain during viral infections and during neuroinflammatory diseases has been previously highlighted [8,9].

Circumventricular organs (CVOs) are specialized brain structures located around the third and fourth ventricles. CVOs differ from the rest of the brain parenchyma in their high vascularization and lack of a proper BBB [10,11]. These specialized areas are points of communication between the blood, the brain parenchyma, and the cerebrospinal fluid. The CVOs are vulnerable to circulating pathogens and might be portals for their entry into the brain. Moreover, CVOs were shown to be involved in pathological conditions such as sepsis, stress and autoimmune encephalitis [12–17]. The *T.b.b.* invasion of the CVOs is one of the first steps of

infection [18–20]. The median eminence (ME), one of the CVOs, adjacent to the arcuate nucleus (Arc) of the hypothalamus has been shown to be particularly susceptible to parasite invasion [18,20].

Despite the importance of T cells in the outcome of *T.b.b.* cerebral infection, an in-depth characterization of the T cell populations and their role in invasion and morbidity is lacking. Moreover, it has not been yet clarified whether T cells can persist after parasiticidal treatment and if they play a role in the neurological consequences of infection. In the present study we show that a majority of T cells in the brain acquired a memory phenotype early after infection and expressed $T_{RM}$ markers at the late stage. T cells and parasites invade the ME/Arc during the early stage of infection, regulating local inflammatory and metabolic responses. During this stage lymphoid cells mediate infection-induced sleep alterations.

## Results

### Increased memory T cell levels in the brain of mice during *T.b.b.* infection

We first characterized the phenotype of T cell populations in the brain after i.p. infection of C57Bl/6 mice with 2000 *T.b.b.* To determine the role of parasites in the maintenance of T cell infiltrates in the brain, a group of mice was treated with the trypanocidal drugs melarsoprol and suramin starting at 21 days post-infection (dpi), when parasites have penetrated into the brain parenchyma. Drug treatment reduced the number of circulating parasites (S1A Fig) and increased the survival of *T.b.b.*-infected mice from 30 dpi for untreated mice, to more than 39 dpi, when treated mice were sacrificed.

The percentage of brain CD45^highCD11b- cells increased at 14 dpi compared to uninfected controls (Fig 1A–1C). The frequency of CD45^highCD11b- cells further augmented at the late stage and after parasiticidal treatment (39 dpi) compared to the early stage of infection (Fig 1B). T cells comprised 80% of the CD45^highCD11b- brain cells (Fig 1C). The drug treatment reduced or eliminated brain parasites but did not affect the levels of T cells in the cortex, corpus callosum and septum of mice (Figs 1D–1F and S1C).

Ninety percent of the CD4 and 60–70% of the CD8 brain T cells showed a CD44+KRLG1- memory phenotype during the early and late stages of infection (Fig 1G and 1H). In comparison, the brain draining deep cervical lymph nodes (dcLN) and the spleen showed lower frequencies of memory T cells (Fig 1I). CD44+ KRLG1+ effector T cells accounted for 3–4% of the CD4 and 11% of the total CD8 T cells in the brain of infected mice (S1D Fig).

Most CD4 and CD8 T cells in the brain showed a CD44+CD62L- effector memory ($T_{EM}$) phenotype. The frequencies of $T_{EM}$ cells in the brain were higher than those in the dcLN, which, conversely, contained higher levels of naïve and central memory T cell ($T_{CM}$) populations. The levels of both CD4 $T_{EM}$ and CD8 $T_{CM}$ in the dcLN were increased after *T.b.b.* infection (Figs 1J and 1K and S1E). The frequency of CD4 but not CD8 dcLN naïve T cells was reduced after *T.b.b.* infection (Fig 1J and 1K).

Resident memory T cells ($T_{RM}$) are a subpopulation of memory T cells that are retained within non-lymphoid tissues including the brain and have a protective or detrimental role against infections or autoimmune diseases [21, 22]. While a relatively low fraction of CD4 T cells were characterized as $T_{RM}$ early after infection, a large percentage expressed $T_{RM}$ cell markers (CD69+CD11a+) during the encephalitic stage of infection (Fig 2A and 2B). CD8 CD69+CD103- T cells were also elevated early after infection, while CD8 $T_{RM}$ cells expressing both CD69 and CD103 were elevated in the late phase of infection (Fig 2C–2E). The frequency of $T_{RM}$ diminished after treatment with melarsoprol and suramin (Fig 2A–2E), despite the high density of brain T cell infiltrates in these animals (S1C Fig). After *T.b.b.* infection, the levels of $T_{RM}$ frequencies were higher in the brain compared to spleens or dcLN (Fig 2F).

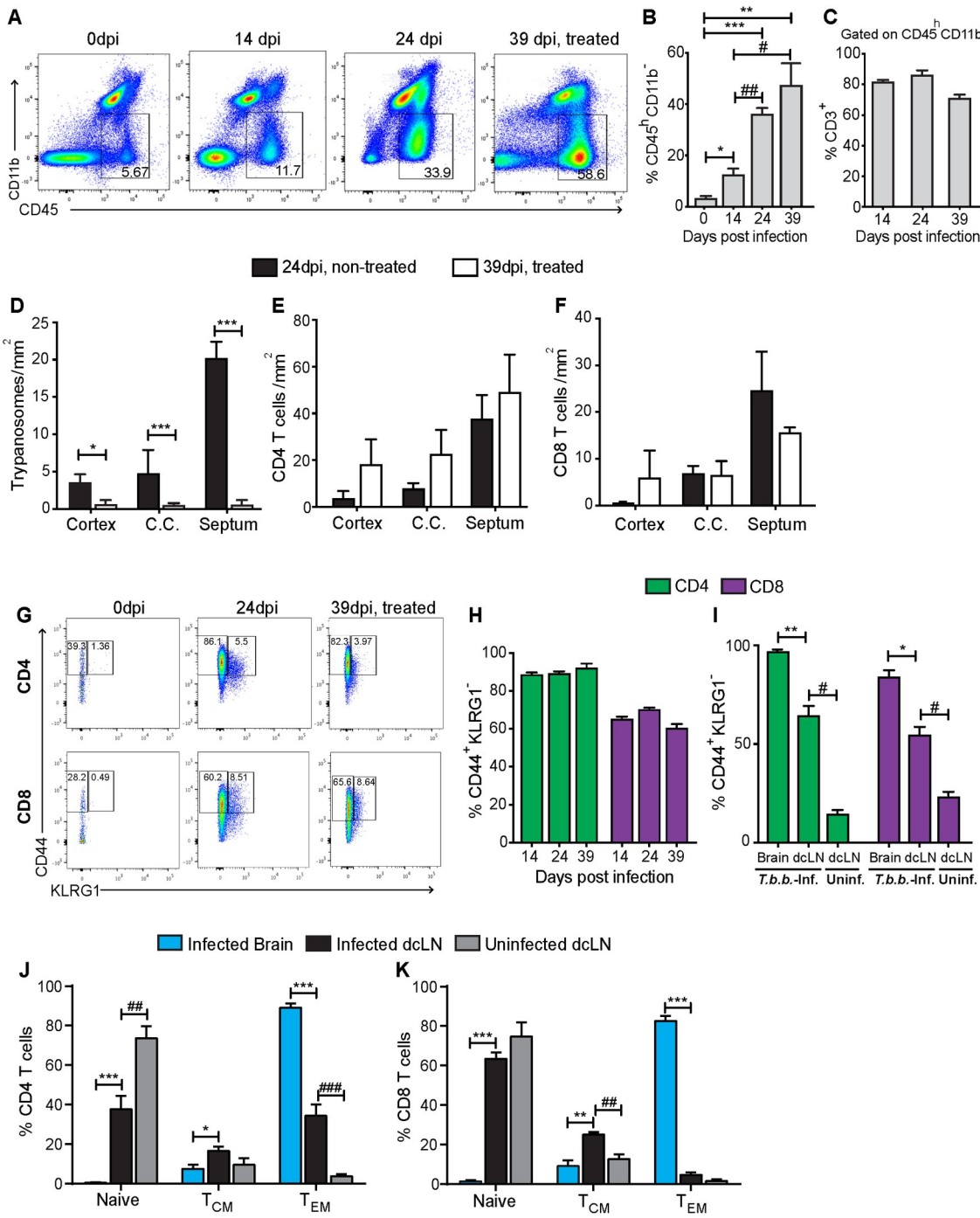

**Fig 1. Phenotypic characterization of brain T cells during infection with *T.b.b.*** (A-C) Mice were infected with *T.b.b.* and sacrificed at the indicated time points. Mice at 39 dpi were treated from day 21 with melarsoprol and suramin. (A) Representative FACS dot plots showing the increase of CD45highCD11b- cells during infection with *T.b.b.* (B-C) The mean percentage of CD45highCD11b- cells in the brain cell suspensions from mice ± SEM at different time points after infection with *T.b.b.* (n = 5 mice per group) (B) and the mean percentage of CD3+ T cells ± SEM within CD45highCD11b- brain populations (C). Differences with uninfected controls are significant at *p≤0.05, **p≤0.01 and ***p<0.001 unpaired Student's t test with Welch's correction. Differences between infected groups are significant at #p≤0.05 and ## p≤0.01 unpaired Student's t test with a Welch's correction. (D-F) The mean number of *T.b. b.* (D), CD4 (E) or CD8 (F) T cells per mm² in different brain regions of mice at 24 or 39 dpi (in the latter group after parasiticidal treatment) is shown. Differences in parasite densities are significant at *p≤0.05 and ***p≤0.001, unpaired Student's t test with Welch's correction. (G) Representative dot plot of CD44 and KLRG1 expression in CD4 and CD8 T cells from the brain of *T.b.b.*-infected mice. (H) The mean % CD44+KRLG1- (memory) CD4 and CD8 T cells ± SEM was evaluated in the brain from mice at

different time points after infection with *T.b.b.* (I) The frequency of memory cells in the brain and deep cervical lymph nodes (dcLN) from *T.b.b.*-infected (24 dpi) or control mice was compared (n = 6 animals per group). (J, K) The mean % $T_{EM}$ (CD44+CD62L-), $T_{CM}$ (CD44+CD62L+) and naïve (CD44-CD62L+) T cells ± SEM gated within CD4 (J) or CD8 (K) T cells in the brain and dcLN from *T.b.b.*-infected mice at 24 dpi (n = 6 mice per group) are depicted. (H-K) Differences are significant with the frequencies in the brain at *p≤0.05, **p≤0.01 and ***0.001, or with the uninfected dcLN at ## p≤0.01 and ### p≤0.001 unpaired Student's t test with a Welch's correction.

Additionally, we assessed the expression of the "programmed cell death protein 1" (PD1), a negative regulator of the immune responses during infection [23]. A relevant fraction of brain CD4 and CD8 T cells expressed PD1 already at 14 dpi (Fig 2G and 2H). Higher levels of PD1 + CD8 memory T cells were determined in the brain during the late than in the early stage of infection (Fig 2H). The levels of PD1 expressing memory CD4 T cells were reduced in parasiticidal-treated mice. Instead, the frequency of PD1+ CD8 $T_{EM}$ was similar before and after treatment with melarsoprol of *T.b.b.* infected mice (Fig 2H). At the late time point, PD1-expressing memory CD4 and CD8 T cells were also increased in the dcLN nodes albeit at lower levels than those in the brains (Fig 2I).

Thus, memory T cells are found in the brain parenchyma early after *T.b.b.* infection, and during the late phase of infection a significant fraction of T cells express $T_{RM}$ markers and PD1.

## *T.b.b.* and leukocytes selectively invade the CVOs of mice early after infection in a lymphoid cell regulated manner

The brain invasion of parasites and leukocytes early and late after *T.b.b.* infection of *rag1*[-/-] mice, lacking mature T and B cells, and WT controls was then compared. *Rag1*[-/-] mice showed higher parasitemia levels than WT mice (S2A Fig) and increased cumulative mortality (p<0.05, $\chi^2$ test), without a significant weight loss (S2B Fig).

*T.b.b.* were detected in the ME of WT mice during an early stage of infection, i.e. 10 dpi (S2C Fig). Parasites were only present within the cerebral blood vessels in both *rag1*[-/-] and WT mice when measured at 14 dpi. The median eminence (ME) was characterized by the expression of vimentin in tanycytes (S2D Fig) [24], the presence of laminin and CD31 (an adhesion receptor) in fenestrated vessels at the base of the ME (S2E and S2F Fig) that were not labelled for glucose transporter 1, a major transporter in the BBB (S2C Fig). *Rag1*[-/-] mice showed higher levels of parasites in the ME (Fig 3A and 3B) and the choroid plexus compared to WT mice during early infection (Fig 3C and 3D). At later time points of infection (23–27 dpi), *T.b.b.* were also observed in the parenchyma of WT (S1B Fig) but not *rag1*[-/-] mice confirming previous studies [3,5].

Clusters of CD45+ leukocytes were observed in the ME of WT mice at 14 dpi but not in uninfected controls (Fig 3E and 3F). Scattered leukocytes were also observed in the Arc (Fig 3E and 3G), and in the brain parenchyma at 14 dpi (Figs 3H and S3A), when parasites were confined within the vessels (S3A Fig). The density of CD45+ cells was lower in the ME of *rag1*[-/-] compared to WT *T.b.b.* infected animals (Fig 3F and 3G). In *rag1*[-/-] mice CD45+ cells were located almost exclusively in the base of the ME (Fig 3E). CD45+ cells were not detected in the brain parenchyma of *rag1*[-/-] mice during the early or late stage of infection (Fig 3H and S3A). Both CD4 and CD8 T cells were present in the ME and some infiltrated the Arc early after infection (Fig 3I–3K). Thus, levels of leukocytes and parasites are detected in the ME of both WT and *rag1*[-/-] mice at 14 dpi. However, leukocytes but not *T.b.b.* were observed in the brain parenchyma of immunocompetent mice at an early stage of infection.

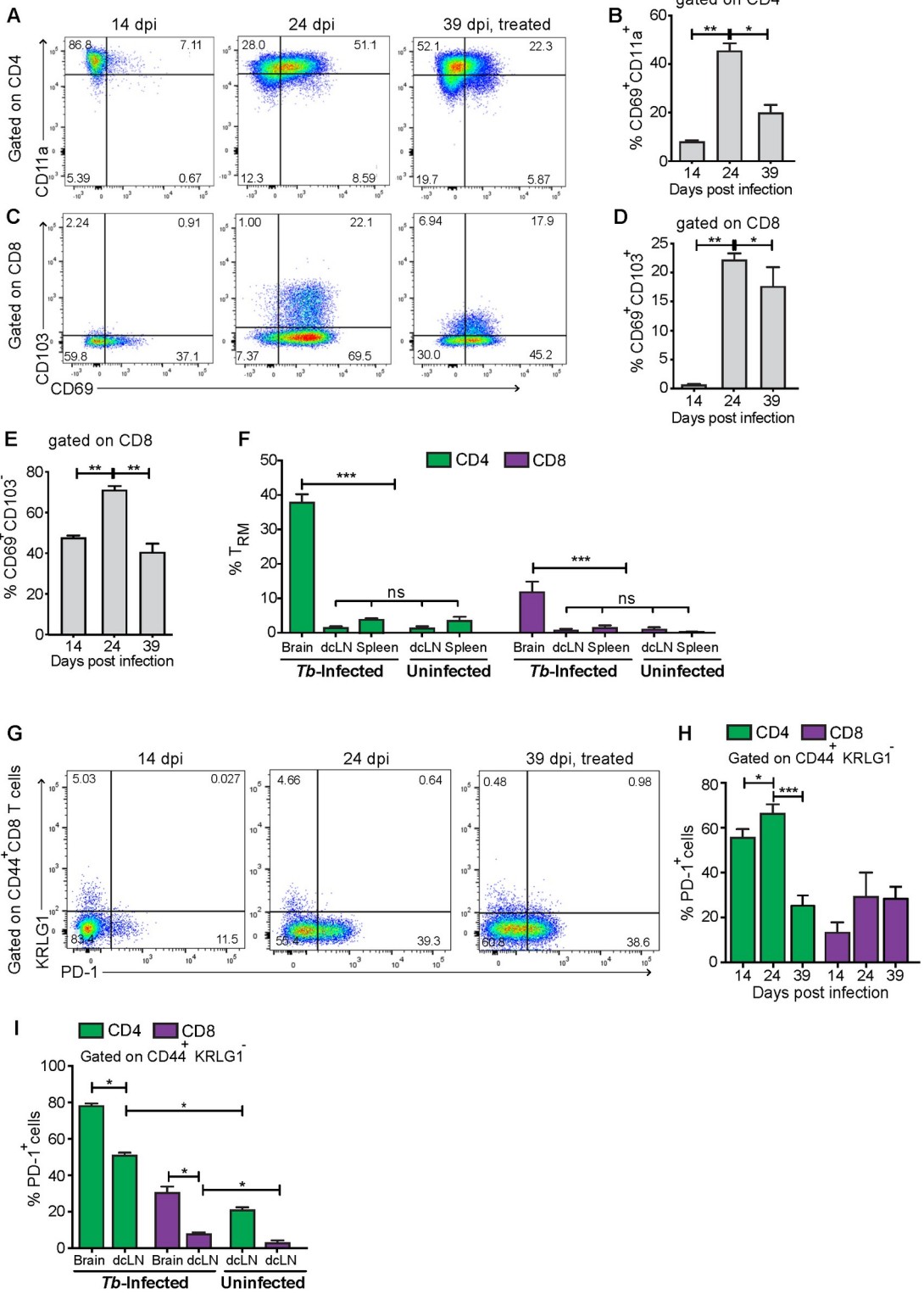

**Fig 2. Increased T_RM frequencies in the brain of mice during the late stage of *T.b.b.* infection.** (A, C) Representative dot plots and (B, D) mean % of CD4 and CD8 T_RM ± SEM in the brain at different times after infection with *T.b.b.* (n = 5 mice per group). The T_RM were gated on CD3+CD44+KLRG1- cells. (E) The mean % of CD69+CD103- CD8 T_RM ± SEM in the brain of mice at different time points after infection with *T.b.b.* is depicted. (B, D, E) Differences are significant between groups at [***] p≤0.01 unpaired Student's t test with Welch's correction (n = 5 mice per group). (F) The mean % CD69+CD11a+ CD4 T_RM and CD69

+CD103+ CD8 T$_{RM}$ ± SEM within CD4 and CD8 T cells in the brain, spleen and dcLN from mice at 24 dpi or control mice are represented. Differences are significant between groups at *** p≤0.001 one way ANOVA. (G) Representative dot plots showing the expression of PD1 and KLRG1 in brain CD8 T cells at 14, 24 and 39 dpi. (H) The mean frequency of CD44+PD1+KRLG1- CD4 and CD8 T cells ± SEM in the brain at different times after *T.b.b.* infection is depicted. (I) The frequency of PD1+ CD4 and CD8 T cells in the brain and dcLN from *T.b.b.*-infected (24 dpi) and uninfected controls are depicted. (H, I) Differences are significant between groups at *p≤0.05, **p≤0.01 and *** p≤0.001 unpaired Student's t test with Welch's correction.

## CD4 and CD8 T cells are redundant for the invasion of *T.b.b.* or the other T cell population into the brain parenchyma

We then tested whether CD4 and CD8 T cells were required for the invasion of leukocytes and parasites into the ME early after infection. In contrast to *rag1$^{-/-}$*, CD4 genomically deficient mice (*cd4$^{-/-}$*) showed similar densities of CD45+ cells and parasites spreading all over the ME and Arc as compared to those in WT mice early after infection (Fig 4A–4D). *Cd4$^{-/-}$* and WT mice also showed similar parasitemia (S3B Fig). The *T.b.b.*, leukocyte and CD8 T cell levels in the brain parenchyma during the late stage of infection of *cd4$^{-/-}$* and WT mice were also similar (Figs 4E and 4F and S3E). Unexpectedly, the levels of *ifng*, *inos* and *tnf* transcripts in the brain of *cd4$^{-/-}$* and WT infected controls were similar (Fig 4G–4I), suggesting that CD4 T cells are redundant for the brain invasion by parasites and other leukocytes.

We then compared the invasion of parasites and T cells into the brain of mice genomically lacking either MHC-I (*mhc1$^{-/-}$*) or β2-microglobulin (*b2m$^{-/-}$*) mice, with impaired thymic maturation of CD8+ thymocytes resulting in the lack peripheral CD8 T cells. *Mhc1$^{-/-}$* and *b2m$^{-/-}$* mice showed similar parasitemia levels as compared to their WT controls (S3C and S3D Fig). Neither *mhc1$^{-/-}$* nor *b2m$^{-/-}$* showed impaired penetration of *T.b.b.*, CD4 T cells or leukocytes into the brain (Figs 4J–4M and S3C). *Mhc1$^{-/-}$* mice showed increased parasite levels in the cortex and corpus callosum as compared to controls, but such differences were not observed between *b2m$^{-/-}$* and WT mice (Fig 4J and 4K). The levels of CD4 T cells and CD45+ leukocytes in the brain parenchyma of either *b2m$^{-/-}$* or *mhc1$^{-/-}$* mice was similar to those in the WT controls (Figs 4L and 4M and S3C). Together this suggests that both CD4 and CD8 T cells invade the brain parenchyma and promote invasion by other leukocytes and parasites during the late stage of *T.b.b.* infection independently of the presence of the other T cell population.

The similar concentration of *ifng*, *inos* and *tnf* mRNA in the brain parenchyma of *T.b.b.*-infected WT and *cd4$^{-/-}$* mice suggested that either CD4 T cells played a minor role in the stimulation of these immune molecules during infection or that a compensation due to the absence of CD4 T cells occurred in the mutant mice. To address these possibilities, we compared the inflammatory molecule levels and parasite levels in the brain of *rag1$^{-/-}$* mice transferred with CD4 T cells with that of non-transferred infected controls. *Rag1$^{-/-}$* mice reconstituted with CD4 T cells showed similar parasitemia titers (S3G Fig), but increased levels of *ifng*, *inos* and *tnf* mRNA as compared to non-transferred infected controls (Fig 4N–4P). CD4 T cell-transferred *rag1$^{-/-}$* mice but not non-transferred controls also showed the presence of *T.b.b.* in the brain parenchyma (Fig 4Q). These data indicate that during *T.b.b.* infection, CD4 T cells are sufficient to induce the penetration of parasites into the brain parenchyma and can stimulate the production of immune molecules required for parasite and T cells brain invasion. Neither CD4 nor CD8 T cells are required for penetration of the other T cell population or of parasites into the brain parenchyma.

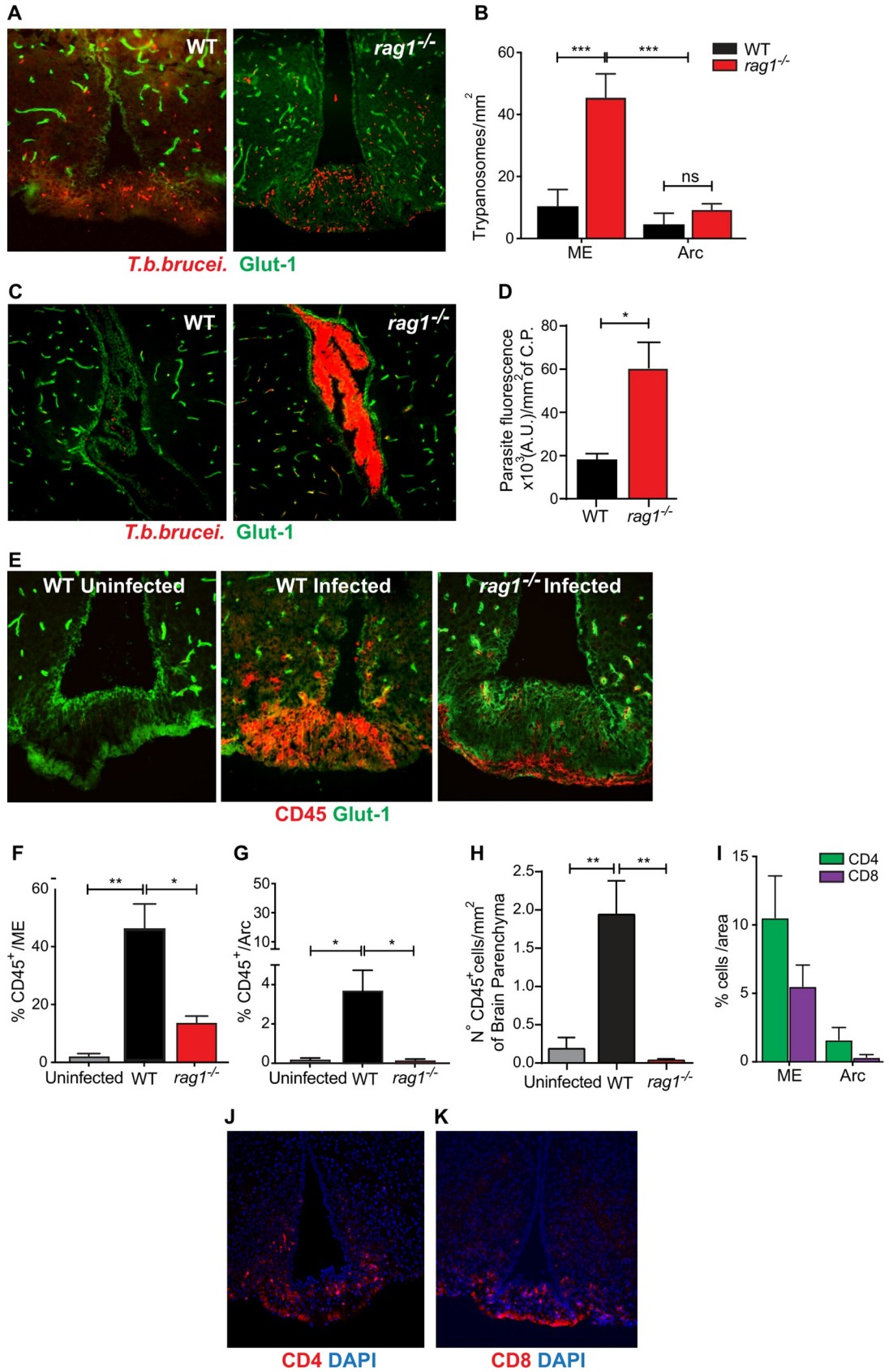

**Fig 3. *T.b.b.* and leukocytes selectively invade the CVOs of mice early after infection in a lymphoid cell regulated manner.** (A) Micrographs showing labelling for *T.b.b.* and Glut1expressing blood vessels in the ME/ Arc and 3rd ventricle area from WT and *rag1*[-/-] mice at 14 dpi. (B) The mean parasite density ± SEM per mm$^2$ in the Arc and ME of WT and *rag1*[-/-] mice (n = 5) at 14 dpi is depicted. Differences between groups are significant at ***p≤0.001 (non-parametric one-way ANOVA Kruskal-Wallis followed by Tukey's test for multiple comparisons). (C) Micrographs showing *T.b.b.* and Glut1 labelling in the choroid plexus of WT and *rag1*[-/-] infected mice at 14 dpi. (D) The mean parasite labelling density in the choroid plexus ± SEM (n = 4). Differences between *rag1*[-/-] and WT mice are significant (*p≤0.05 Mann-Whitney U test). (E) Micrographs showing staining for CD45 and Glut1 in the ME/ Arc from WT or *rag1*[-/-] mice at 0 or 14 dpi. Note the dissimilar distribution of CD45+ cells in the ME and Arc of WT and *rag1*[-/-] *T.b.b.*-infected mice. (F, G) The mean % area with CD45+ cells in the ME (F) and Arc (G) ± SEM (n = 5 mice per group) is depicted. Differences between groups are significant (*p≤0.05 and ** p≤0.01 Mann-Whitney U test). (H) The mean numbers of CD45+ cells in the brain parenchyma of mice 14 dpi with *T.b.b.* per mm$^2$. Differences with WT-infected mice are significant (**p≤0.01 Mann-Whitney U test). (I) The mean % area with CD4+ or CD8+ cells in the ME and Arc ± SEM is represented. (J, K) CD4 (J) and CD8 (K) and DAPI (nuclear) staining in the ME and Arc of WT mice at 14 dpi.

## Increased lymphoid cell dependent and independent expression of inflammatory and metabolic genes in the ME/ lateral hypothalamus of *T.b. b.* infected mice

The levels of transcripts coding for immune and metabolic molecules were then measured. MECA-32 is a stress-regulated antigen expressed in the fenestrated capillaries of CVOs but not in the BBB [25]. Increased *meca32* transcripts levels were observed in the ME and lateral hypothalamus (including Arc) of *rag1*[-/-] infected animals (Fig 5A). The expression of vascular-endothelial growth factor-α (VEGF-A) in the ME of mice has been shown to increase hypothalamus BBB permeability [26]. The expression of *vegfa* and *glut1* mRNA are both triggered by the hypoxia-inducible factor-1α (HIF-1α) transcription factor [26–29]. Interleukin (IL)-1β, another HIF-1α regulated molecule [30,31], has been shown to control the outcome of infection with *T.b.b.* [32]. The levels of *vegfa*, *glut1* and *il1b* mRNA were found to be increased in the ME/ hypothalamus (but not in the cortex) of both *rag1*[-/-] and WT mice early after infection, (Fig 5B–5D). *Hif1a* mRNA levels were also elevated in the ME/ hypothalamus after parasite infection (Fig 5E). Thus, our results suggest that HIF-1α controls gene expression within the ME/hypothalamus in a T-cell-independent manner.

The expression of *ifng* mRNA in the ME/ hypothalamus and the cortex was compared at different time points after infection. *Ifng* mRNA was increased in the ME/hypothalamus of WT mice at 16 and 27 dpi as compared to uninfected mice. Instead, levels of *ifng* mRNA in the ME/hypothalamus of *rag1*[-/-] mice were not increased (Fig 5F). The levels of *ifng* mRNA in the cortex were increased in the encephalitic stage of infection (Fig 5F). *Tnf* mRNA levels increased in the ME/hypothalamus of both *rag1*[-/-] and WT mice at 16 dpi. In the cortex *tnf* mRNA levels were increased at late times after infection as compared to uninfected controls (Fig 5G).

In summary, infection with *T.b.b.* causes structural, immunological and metabolic changes in the ME/ Arc, which are in part regulated by lymphoid cells.

## Sleep episodes are reduced in *T.b.b.* infected *rag1*[-/-] mice

Sleep alterations are common during brain infection [33] and have been reported in rats during the early stages of *T.b.b.* infection using electroencephalography/ electromyography recordings [34]. Here, sleep and wake states were determined by videorecording with infrared cameras, to avoid possible tissue damage and inflammation due to the implant of electrodes for polygraphic assessment. It has been previously found that video-monitoring consistently estimates total sleep time with high accuracy [35,36]. Motor activity was quantified by custom-made video-based motion detection algorithms. The sleep-wake pattern of WT and *rag1*[-/-] *T.b. b.*-infected mice at 13 dpi were recorded at the beginning of both light (Zeitgeber time, ZT

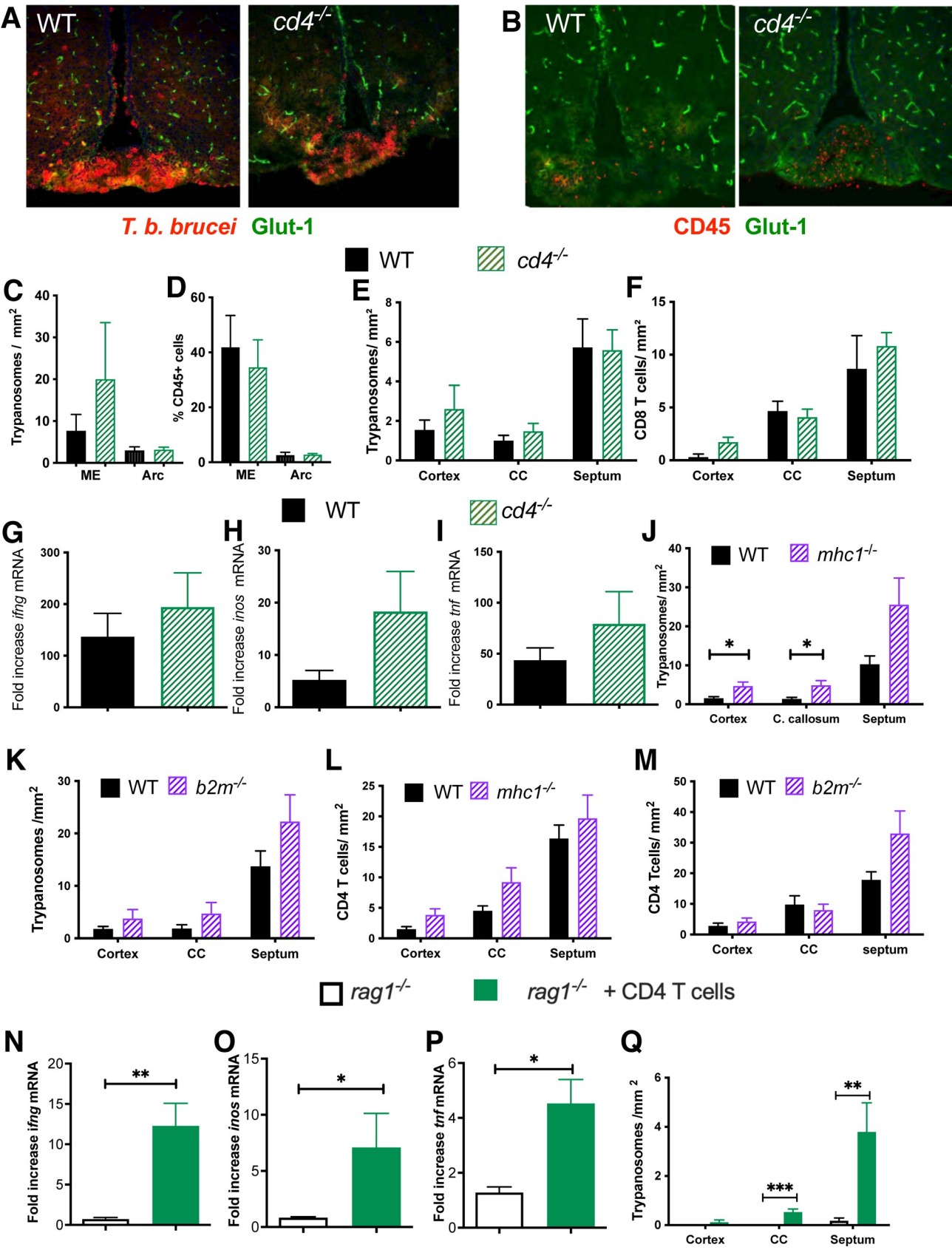

**Fig 4. Neither CD4 nor CD8 T cells are required for penetration of the other T cell population or of parasites into the brain parenchyma.** (A-D) Micrographs showing labelling for *T.b.b.* (A) or CD45+ cells (B) and Glut1 expressing blood vessels in the ME/ Arc and 3$^{rd}$ ventricle area from WT and *cd4*$^{-/-}$ mice at 14 dpi. The mean *T.b.b.* (C) or CD45+ cell (D) density ± SEM per mm$^2$ in the Arc and ME of WT and *cd4*$^{-/-}$ mice (n = 5) at 14 dpi is depicted. (E, F) The mean number of *T.b.b.* (E) and CD8 T cells (F) per mm$^2$ in different brain regions of *cd4*$^{-/-}$ and WT at 23 dpi is shown. (G-I) The mean fold increase of *ifng* (G), *inos* (H) and *tnf* (I) mRNA levels ± SEM was measured by real-time PCR in total RNA from the brain at 23 dpi after *T.b.b.* infection of WT and *cd4*$^{-/-}$ mice (n≥5 per group). The relative concentration of these transcripts was normalized to those from non-infected WT mice. (J-M) The mean number of *T.b.b.* (J, K) and CD4 T cells (L, M) cells per mm$^2$ in different brain regions of *mhc1*$^{-/-}$ (J, L) and *b2m*$^{-/-}$ (K, M) as compared to those WT mice at 23 dpi is shown (n = 5 per mouse). Differences in *T.b.b.* density between *mhc1*$^{-/-}$ and WT mice were significant (*p≤0.05 Mann-Whitney's U test). (N-Q) *Rag1*$^{-/-}$ mice were administered i.v. with 10$^6$ FACS sorted CD4 T cells and infected with *T.b.b.* 3 weeks after transfer. The mean fold increase in *ifng* (N), *inos* (O) and *tnf* (P) mRNA levels ± SEM was measured by real time PCR in total RNA from the brain parenchyma of either *rag1*$^{-/-}$ mice transferred with CD4 T cells or non-transferred *rag1*$^{-/-}$ controls at 20 days after infection (n≥4 per group), and were normalized to uninfected brains from *rag1*$^{-/-}$ mice. Differences in mRNA levels in the between CD4 transferred and non-transferred *rag1*$^{-/-}$ mice were significant (*p≤0.05 and **p≤0.01 Mann-Whitney's U test). (Q) The mean number of *T.b.b.* per mm$^2$ ± SEM in different brain regions of *rag1*$^{-/-}$ transferred with CD4 T cells and non-transferred *rag1*$^{-/-}$ controls at 20 dpi is shown (n = 5 per group). Differences in *T.b.b.* density between CD4 transferred and non-transferred *rag1*$^{-/-}$ mice were significant (**p≤0.01 and ***p≤0.001 Mann-Whitney's U test).

0–3) and dark period (ZT 12–15) of the light-dark cycle. We observed a reduction of the mean duration of sleep episodes in WT-infected compared to *rag1*$^{-/-}$ -infected mice at the beginning of the light period (ZT 0–3) when rodents spend most time asleep (Fig 5A–5D). As a consequence of this reduction, WT-infected mice showed an increase in the number of sleep episodes and of transitions between vigilance states, an index of sleep fragmentation [37], compared to *rag1*$^{-/-}$ infected mice (Fig 6A–6E). Thus, lymphoid cells may regulate the sleep alterations during infection with *T.b.b.*

## Discussion

Although T cells have been previously shown to be required for *T.b.b.* invasion into the brain, a detailed phenotypic and functional characterization of T cells in the outcome of *T.b.b.* infection has not been performed.

We found that T cells in the brain increased during infection and persisted after reducing *T.b.b.* levels by parasiticidal treatment, recapitulating the post-treatment reactive encephalopathy occurring after a sub-curative parasiticidal treatment of the late stage of *T.b.b.* infection. Most brain T cells showed a T$_{EM}$ phenotype (Cd44+CD62L-KLRG1-) with low effector functions but with the capacity to proliferate since the early stage of infection. The frequencies of CD4 and CD8 T$_{RM}$ cells increased in the brain during the late stage of infection. The brain T$_{RM}$ cells also constituted a major T cell population after parasiticidal treatment. T$_{RM}$ cells develop from KLRG1- T$_{RM}$ precursor cells [38] that either selectively infiltrate the brain via the CVOs during the early stage of the *T.b.b.* infection or transverse the BBB attracted by CXCL9 and CXCL10 during the late stage of infection. CXCL9 and CXL10 are highly expressed during the late stage of *T.b.b.* infection in an IFN-γ-dependent manner [39]. A fraction of CD8 T cells in the brain during the late stage of brain infection expressed CD103, an integrin that binds to E cadherin, together with CD69, markers that define CD8 T$_{RM}$. In the adult mouse brain, E-cadherin expression is limited to cells lining the ventricles [40], while T$_{RM}$ cells have been found all through the parenchyma, suggesting that CD103 may be binding to unique ligand(s) within the CNS [41].

Expression of CD103 seems to occur in some but not all T$_{RM}$. CD103-CD69+T$_{RM}$ cells have been described in secondary lymphoid organs [42], in the gut [43], in the female genital tract [44] and in the brain [8]. During the early stage of infection, 40–50% of brain CD8 T cells were CD69+CD103-.

The expression of PD-1 on T cells hampers terminal effector responses and promotes the survival of memory cells. PD-1 expressing memory cells are proliferative in contrast to KLRG-1+ cells that show a low life span, a high level of cytokine secretion [45] and parenchymal

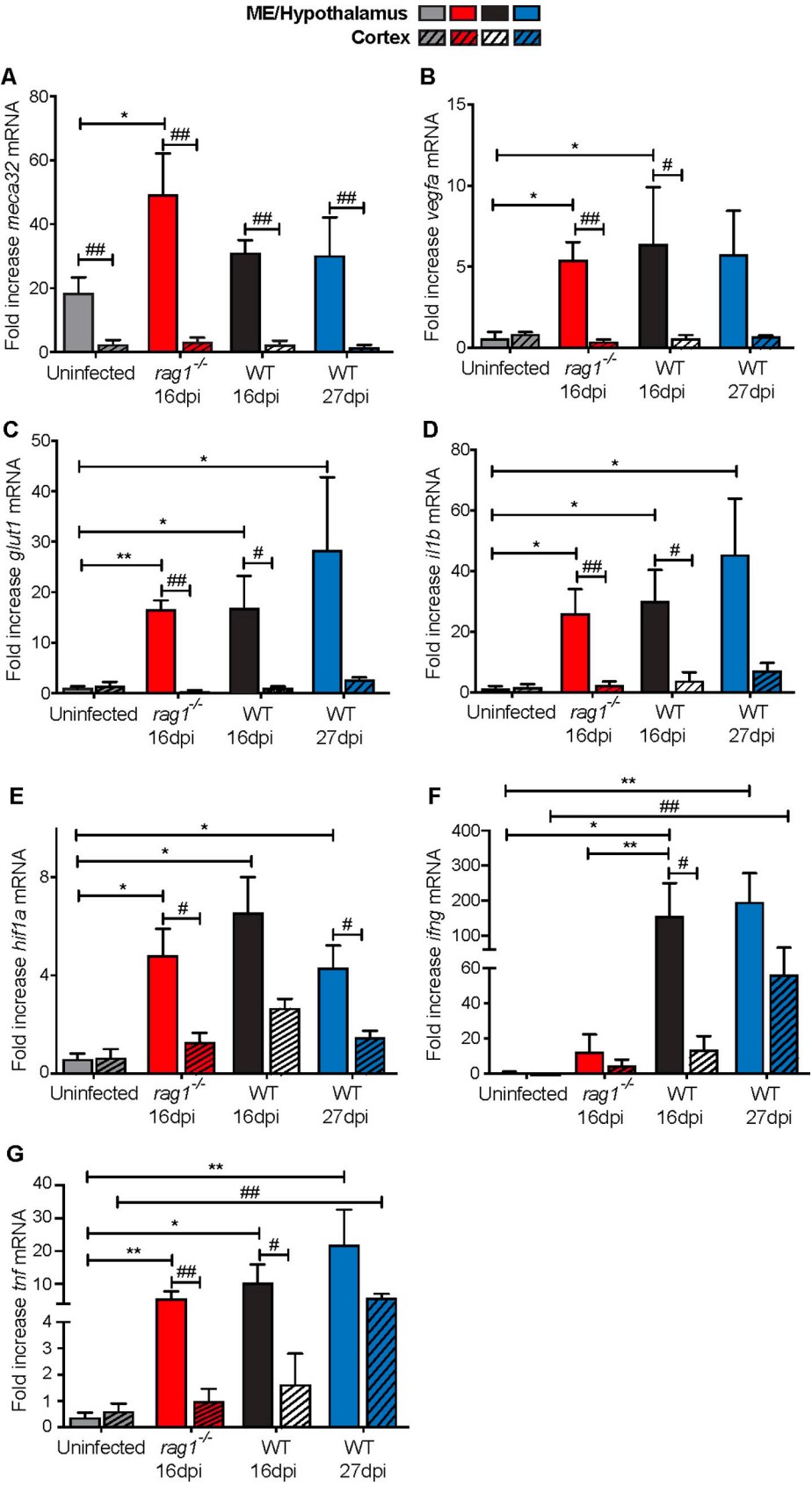

**Fig 5. Increased lymphoid cell dependent and independent expression of inflammatory and metabolic genes in the ME/ lateral hypothalamus of *T.b.b.* infected mice.** (A-D) The mean fold increase in (A) *meca32*, (B) *vegfa*, (C) *glut1*, (D) *il1b*, (E) *hif1a*, (F) *ifng, and* (G) *tnf* mRNA levels ± SEM was measured by real time PCR in total RNA from either cortex or ME/ hypothalamus obtained before or at different times after *T.b.b.* infection of WT and *rag1*⁻/⁻ mice (n≥4 per group). Differences between cortex and ME/ hypothalamus in the same group of animals are significant (#p<0.05 and ## p<0.01 Mann-Whitney's U test). Differences in mRNA levels in the ME/ hypothalamus between groups are significant (*p≤0.05 and **p≤0.01 Mann-Whitney's U test). Differences in mRNA levels in the cortex between groups are significant (# p≤0.05 and ## p≤0.01 Mann-Whitney's U test).

localization [46]. Rather than exhausted, PD-1 expressing cells are maintained upon chronic infections, with persisting epitope targets [23]. Already at 14 dpi a fraction of brain T cells expressed substantial levels of PD-1, the frequencies and numbers of PD1 expressing T cells increased during the late stage of infection. Mice showed reduced frequencies of $T_{RM}$ and PD1 + memory CD4 T cells in the brain after treatment with melarsoprol.

*T.b.b.* have been previously found in the ME/Arc during the early stage of infection [19]. We observed parasite and leukocyte invasion of the ME/Arc in both WT and *rag1*⁻/⁻ mice early after infection. The parasite number was higher in the ME of *rag1*⁻/⁻ mice, while fewer CD45 + cells mainly located in the base of the ME were found in *rag1*⁻/⁻ mice and none was observed in the brain parenchyma suggesting a role for lymphocytes in the invasion. CD4 T cells proved redundant for invasion into the ME and Arc. Both CD4 and CD8 T cells may, independently of each other, trigger the penetration of *T.b.b.* and leukocytes into the brain parenchyma during the late stage of infection. We found that the levels of *ifng, tnf* and *inos* mRNA in the brain was not reduced in *cd4*⁻/⁻ *T.b.b.*-infected mice. This is likely due to a compensation by other cell types in the stimulation of these immune molecules by CD4 T cells, since we observed *rag1*⁻/⁻ mice reconstituted with CD4 T cells showed an increased production of these transcripts in the brain after *T.b.b.* infection. Indeed, the numbers of CD8 were normal in the thymus but were expanded in the periphery. Moreover, MHC-II-restricted, double negative TCRαβ⁺ CD4-CD8-T cells have been shown to develop in *cd4*⁻/⁻ mice [47], and the CD8 T cell population was shown to be contained with MHC-II restricted cells [48]. Despite these compensations, T cell help for the production of antibodies [49], and CD8 T cell cytotoxicity were reduced in cd4⁻/⁻ mice [50]. *Cd4*⁻/⁻ mice have also been shown to be susceptible to experimental parasitic, bacterial and viral infections [50–52]. We also found that CD4 T cells were sufficient to mediate the penetration of parasites into the brain.

The ME as well as other CVOs are highly vascularized and possess small fenestrations that allow diffusion of small molecules. Tight junction proteins between specialized tanycytes that outline the ME will instead prevent the free passage of molecules from the ventricle lumen into the Arc [8]. Our results also show that *T.b.b.* infection alters the metabolic status and triggers the penetration of leukocytes into the ME. *Hif1a* transcripts and HIF-1-regulated genes coding for inflammatory and metabolic mediators were increased in the ME/Arc area. The penetration of cells and parasites during the infection might be also related to structural changes associated with the expression of MECA32 that has been previously described to alter the permeability of the blood-hypothalamus barrier [26]. In this regard, VEGF-α produced by tanycytes or by MCH neurons has been shown to enhance the permeability of the ME [26, 53]. We observed that the levels of *vegfa* and *meca32* mRNA was increased in the ME/Arc during *T.b.b.* infection. Additionally, *T.b.b.* infection increased the levels of *il1b* and *tnf* mRNA in the ME/Arc. IL-1β and TNF activate endothelial cells, resulting in increased expression of adhesion molecules, metalloproteases and chemokines that mediate increased permeability and leukocyte extravasation. Furthermore, *ifng* mRNA expression increased in the ME/Arc during the early stage of *T.b.b.* infection. IFN-γ is required for astrocyte expression of CXCL10 chemokine, which is in turn required for the invasion of CXCR3+ Th1 cells into the brain [6, 54].

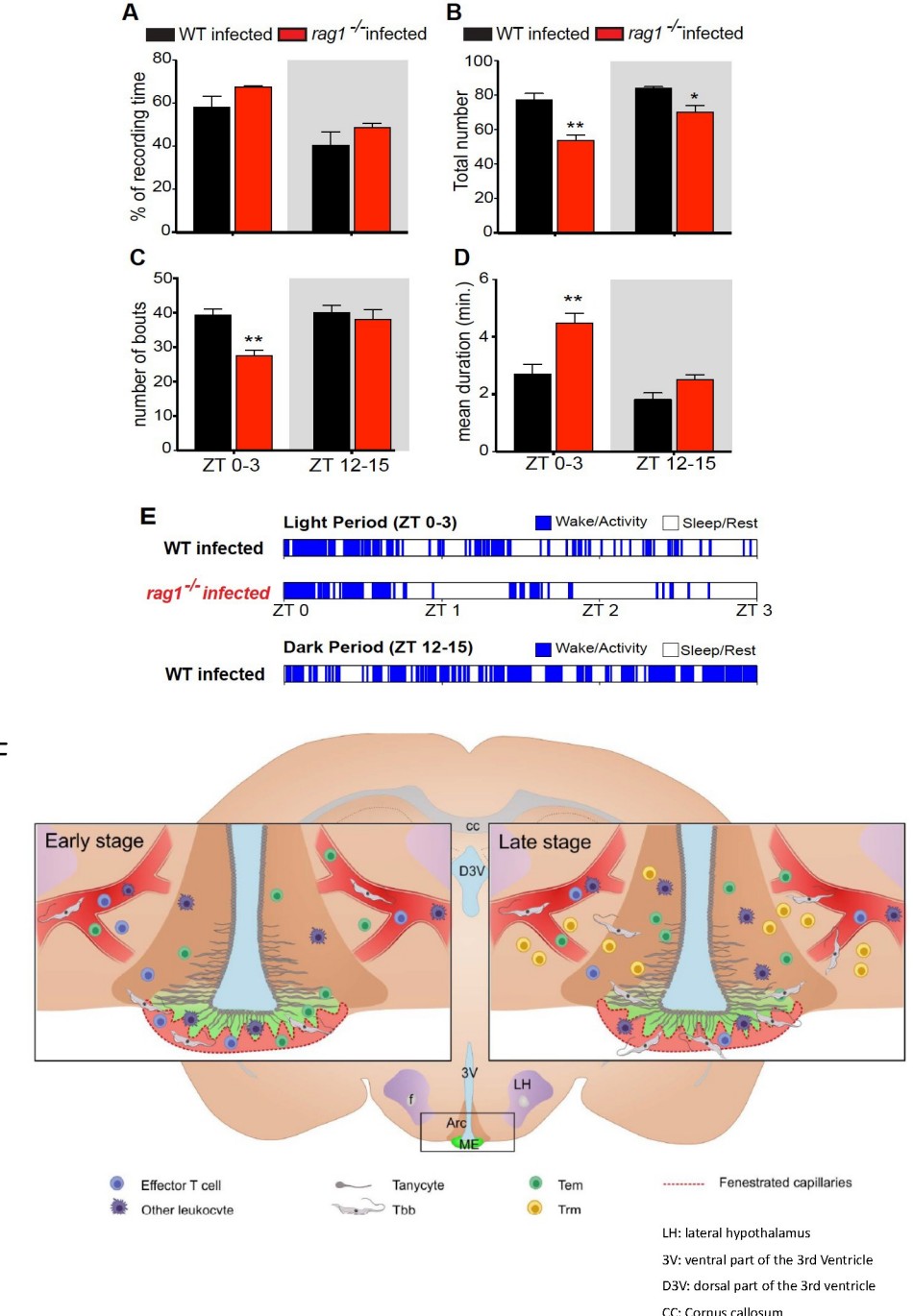

**Fig 6. Lymphoid cells mediate the sleep alterations during _T.b.b._ infection.** (A) Percentages of time spent in sleep phase, (B) the total number of sleep/wake transitions, (C) the total number of sleep bouts (episodes) and (D) the mean duration (in minutes) of sleep episodes in WT (in black) and _rag1_[-/-] (in red) _T.b.b._-infected mice (n = 4 mice per group) at 13–14 dpi. Data are mean ± SEM and are obtained at two different time points: Zeitgeber time (ZT) 0–3 (light period) and ZT 12–15 (dark period). Grey areas represent the dark period of the light-dark cycle. Differences between _rag1_[-/-] and WT mice are significant (*$p \leq 0.05$, ** $p \leq 0.01$ Student's t-test). (E) Representative alternating patterns of sleep (in white) and wake episodes (in blue) in one WT and one _rag1_[-/-] _T.b.b._-infected mouse assessed using digital video analysis. (F) _Graphical abstract_: Early after infection, _T.b.b._ and leukocytes will invade the ME (and other CVOs) that are irrigated by fenestrated vessels lacking a BBB. In the ME the barrier has been displaced to the walls lining the 3rd ventricle and the Arc, where tanycytes, specialized radial glia cells form a physical barrier to control the correct transport of nutrients and metabolic hormones into the brain parenchyma. During the early systemic infection,

*T.b.b.* parasites and leukocytes migrate from the vascular layer at the bottom of the ME into the floor of the 3rd ventricle and into the Arc. We show that T cells facilitate the passage of leukocytes into the lumen of the ME and the Arc but hamper parasite invasion. T cells, other inflammatory cells as well as *T.b.b.* stimulate the production of metabolic and inflammatory molecules that increase the permeability of blood hypothalamus barrier. Further T cell activity influence sleep wake rhythm during *T.b.b.* infection. At this early-stage T cells in the brain differentiate into a memory phenotype ($T_{EM}$), and low levels of these can be found in the brain parenchyma. In a second stage of infection CD4 and CD8 T cells and *T.b.b.* invade the brain parenchyma by breaching the BBB. A fraction of T cells in the brain will also differentiate into resident memory phenotype ($T_{RM}$). We speculate that $T_{RM}$ secrete chemokines that attract vascular T cells to the brain parenchyma. The graphical abstract was generated with Biorender software.

While IFN-γ is mediated by T cells, T cells were not required for expression of *tnf* and *il1b* mRNA in the ME/Arc during *T.b.b.* infection.

The susceptibility of ME/ Arc to early *T.b.b.*-infection and the local inflammatory response might be associated with the pathogenesis of African trypanosomiasis. Here we showed that lymphocytes can regulate the dynamics of the sleep-wake cycle at early stages after *T.b.b.* infection. While total sleep time did not differ between groups, infected WT mice showed an increased number of sleep/wake phase transitions and a reduced sleep episode duration as compared to those shown by *rag1*$^{-/-}$ infected mice. Distinct sleep changes during *T.b.b.* infection can be due to altered function or damage of the population of neurons that reside in the lateral hypothalamus which releases the orexin peptide pair [33,55–57]. A role for T cells in narcolepsy, where sleep dysregulation resembles the one of *T.b.* infection, has also been implied by the presence of autoreactive orexin and other orexin-neuron antigen-specific CD4 and CD8 T cells, by the strong HLA association and by the presence of a skewed T cell receptor repertoire in narcolepsy patients [58–61].

In summary here we reported that: 1) most T cells in the brain show a memory phenotype already during the early stage of infection, while high levels of $T_{RM}$ expressing the immunoregulatory PD-1 molecule are observed during the encephalitic stage of infection (Fig 6F); 2) *T.b.b.* parasites and T cells penetrate into the ME and Arc during the first stage of infection (Fig 6F), when lymphoid cells hamper parasite invasion but facilitate that of other leukocytes; 3) scattered leukocytes (but not parasites) are present in the brain parenchyma of WT mice during the early stage of infection; 4) *T.b.b.* infection triggers T cell-dependent and independent immune and metabolic responses in the Arc and ME; 5) neither CD4 nor CD8 T cells are required for the penetration of parasites and leukocytes into the brain parenchyma during the late stage of infection and 6) T cells mediate alterations of the sleep-wake dynamics early after *T.b.b.* infection.

## Methods

### Ethics statement

The animals were housed and handled at the Department of Microbiology, Tumor and Cell Biology and the Astrid Fagreus Laboratory, Karolinska Institutet, Stockholm, according to directives and guidelines of the Swedish Board of Agriculture, the Swedish Animal Protection Agency, and the Karolinska Institute (djurskyddslagen 1988:534; djurskyddsförordningen 1988:539; djurskyddsmyndigheten DFS 2004:4). The study was performed under the approval of the Stockholm North Ethical Committee on Animal Experiments permits number 187/ 12 and 35/15. Animals were housed under specific pathogen-free conditions.

### Mice and parasites

*Rag1*$^{-/-}$, *cd4*$^{-/-}$, *beta-2 microglubulin* (*b2m*$^{-/-}$) and *major histocompatibility complex 1* (*mhc1*$^{-/-}$) mice were generated by homologous recombination in embryonic stem cells. All strains were backcrossed on a C57BL/6 which were used as wild type (WT) controls.

Mice (6–8 weeks old) were infected by intraperitoneal (i.p.) injection with $2 \times 10^3$ parasites of a pleomorphic stabilate of *T.b. brucei*, AnTat 1.1E (obtained from ITG, Antwerp, Belgium). Parasitemia levels were determined by microscopy counting of tail vein blood and body weight monitored during the infection.

When indicated mice were also treated with melarsoprol i.v. and suramin i.p. starting 21 dpi after infection. Suramin (20 mg/ kg body weight) was administered once while melarsoprol 2 mg/ kg in cyclodextrin was given every other day in 6 doses.

## Immunohistochemistry

To determine the presence of parasites and leukocytes in the ME/ Arc 14 μm coronal sections of fresh frozen brains at levels including the septal nuclei and the hypothalamus were cut, mounted and fixed in 4% formalin with 0.17% picric acid in PBS followed by acetone. The sections were double-labeled by incubation with anti-*T.b.b.* VSG, anti-CD45, anti-Glut-1, anti CD4, anti-CD8, anti-CD31, anti ZO-1, anti-vimentin and anti-MECA32 antibodies (S1 Table) followed by the fluorochrome labelled secondary antibodies rhodamine red donkey anti-rat IgG or anti-rabbit IgG; Alexa Fluor 488 donkey anti-goat IgG or anti-rat IgG (all from Jackson Immunoresearch, West Grove, PA) and Alexa Fluor 488 chicken anti-rabbit (InVitrogen). Sections were examined in a Leica DMRE fluorescence microscope. The trypanosome/ mm$^2$ or in a defined brain region were quantified. Parasite fluorescence in the choroid plexus (CP) was quantified by a Zeiss ZEN software microscope (Observer Z1, Zeiss). The CP area was determined by using a scale-calibrated measurement tool and the mean fluorescence intensity of the red channel was obtained in the defined area. When indicated, the fraction of the brain areas labeled with CD45, CD4 and CD8 were also quantified using a Cell Profiler software pipeline.

## Real-Time PCR

Brains were dissected, frozen on dry ice and sectioned in a cryostat. Coronal 50–100 μm cryo sections from the ME/ hypothalamus area were placed on gelatin coated coverslips and were kept on dry ice until microdissection. An Integra Miltex punch 1.5 mm diam was used to extract biopsies from three consecutive sections covering the whole ME/ Arc region which were pooled. Biopsies from the cortex were also obtained. Total RNA was extracted from brain samples using the RNeasy Mini kit (Qiagen) following the manufacturers recommendations. cDNA was obtained by reverse transcription. Transcripts were quantified by real-time PCR as described [62] using primer sequences indicated in S2 Table. *Hprt* was used as a control gene to calculate the ΔCt values for individual brain samples. The relative amount of target transcript/ *hprt* transcripts was calculated using the $2^{-(\Delta\Delta Ct)}$ method [62]. These values were then used to calculate the relative expression of cytokine mRNA in uninfected and infected tissue biopsies.

## Flow cytometry and cell sorting

To prepare single-cell suspensions, brains were dissected, cut into small pieces and incubated with 1mg/ ml collagenase IV (Gibco) and 10 μg/ml DNase I (Roche), for 1h at 37˚C. The cell suspensions were then washed and resuspended in 0.5% FBS, 2mM EDTA PBS (FACS buffer), centrifuged for 10 min and the pellets were mixed with 5 ml of 30% isotonic Percoll solution. After centrifugation at 300 g for 30 min, the myelin (top layer) was removed, the cells were filtered in 40-μm pore size cell strainers and resuspended in FACS buffer and stained as described below.

Spleens and deep cervical lymph nodes (dcLN) were dissected and homogenized on 70 μm-cell strainers. Red blood cells were lysed using buffered 0.8% $NH_4Cl$.

Freshly prepared cell suspensions were washed, incubated with Fc block (BD Biosciences) and stained with CD3, CD11b, CD45, CD3, CD4, CD8, CD44, CD62L, PD1, KLRG1, CD11a, CD69 and CD103 using fluorochrome labelled antibodies (all from eBioscience, San Diego, CA) and fixed with paraformaldehyde before the acquisition. Data were acquired on a LSRII flow cytometer (BD Biosciences) and analyzed with FlowJo software (Tree star Inc., Ashland, OR).

In order to sort CD4 T cells, spleen cells suspensions from uninfected mice were prepared as described above and enriched in T cells using mouse CD90.2 microBeads (MACS) as described by the manufacturer. T cells were then incubated with CD3 and CD4 antibodies, washed, filtered through a 70 μm cell strainer and CD4 T cells were separated by FACS sorting by FACS sorting (MoFlo XDP, Beckman Coulter, Brea, CA).

### Non-invasive assessment of sleep-wake cycle using video analysis

Mice were individually housed and exposed to standard 12h:12h L:D cycles. Under these conditions light onset was designated Zeitgeber time (ZT) 0 and dark onset as ZT12. Ambient temperature was maintained at 22 ± 2˚C and food and water were available *ad libitum*. A video recording combined with video-based motion detection algorithms was used to evaluate periods of sustained immobility as a surrogate of polysomnography [63,64]. Miniature near-infrared (NIR) video cameras (Sentient Mini-night vision CCTV camera, Maplin, UK) were mounted above each cage to permit the recording of mice under either light or dark (NIR) conditions. Cameras were fitted with a wide-angle lens to enable the whole cage area to be viewed. Cages were positioned approximately 30 cm below the camera which was centred over the area containing the mouse. Video-recordings were obtained during the first three hours of both light and dark periods of the light-dark cycle. Video data was extracted from the hard drive recorder and analyzed offline using a custom-made video-based motion detection algorithm written in Matlab (MathWorks) with a time resolution of 1 s. Briefly, the animal motion was detected (1 Hz) by calculating the numbers of pixels whose intensity changed over time (in a similar manner as reported in [35]).

### Statistical methods

The statistical tests were performed using the Prism software (GraphPad, La Jolla, CA). Differences in *T.b.b.* and immune cell densities in tissues, in the expression levels of surface markers and transcript levels in tissue lysates were calculated using the non-parametric Mann-Whitney's U-test. Differences in the total sleep within the recording time, the duration of sleep episodes and the numbers of sleep-wake transitions were calculated by Student's t-test. Statistical differences in the frequencies of cell populations as recorded by FACS analysis were estimated using an unpaired Student's t-test with Welch's correction.

### Supporting information

**S1 Fig. Frequency of T cell memory subpopulations in the brain and dcLN of *T.b.b.* infected mice.** (A) The parasitemia of mice infected with *T.b.b.* and treated with melarsoprol and suramine starting at 21 dpi, is shown. (B) Micrographs comparing the presence of *T.b.b.* in the septum and cortex of untreated and suramin and melarsoprol treated mice at the indicated time points after infection. (C) Micrographs comparing the density of CD45+ leukocytes in the corpus callosum of mice infected or not with *T.b.b.* and treated with melarsoprol and suramin or untreated controls. (D) The mean % CD44+KRLG1+ CD4 and CD8 T cells ± SEM was measured in the brain and dcLN from either *T.b.b.*-infected (24 dpi) or control mice. Differences are significant at * p≤0.05 and ***p≤0.001 Student's t test with Welch's correction.

(E) Representative dot plots showing the expression of CD44 and CD62L in the brain and dcLN CD4 and CD8 T cells from *T.b.b.*-infected and uninfected controls.
(TIFF)

**S2 Fig. Early parasite invasion of the ME.** (A) The mean $\log_{10}$ parasitemia and (B) the percentage of weight change of WT and *rag1*[-/-] mice infected i.p. with 2000 *T.b.b.* (n $\geq$5 animals per group). (C) Micrographs showing the labeling of *T.b.b.* and glut1 in the ME of mice at different days after infection. Parasites were not detected in the ME of mice at 7 dpi. (D) Vimentin staining of tanycytes and other ependymal cells in the ME/ Arc. (E) Micrograph showing the extravascular localization of parasites in the ME brains of mice infected with *T.b.b.* at 14 dpi by staining with CD31. (F) Labeling of laminin and *T.b.b.* showing extravascular parasites in the ME.
(TIFF)

**S3 Fig. Invasion of the brain parenchyma by T cells and parasites during the early stage of T.b.b. infection.** Micrographs showing the labeling for CD45 and glut1 in the cortex of WT and *rag1*[-/-] mice at 14 dpi with *T.b.b.* (B-D) The mean $\log_{10}$ parasites/ ml blood of WT and *cd4*[-/-] (B), *b2m*[-/-] (C) and *mhc1*[-/-] (D) and WT mice at different times after infection with *T.b.b.* (n $\geq$5 animals per group). (E, F) The mean number of CD45+ cells per $mm^2$ in different brain regions of *cd4*[-/] (E), *b2m*[-/-] (F) and WT mice at 23 dpi is shown. (G) The mean $\log_{10}$ parasites/ ml blood of CD4 T cell transferred and non-transferred *rag1*[-/-] mice at different times after infection with *T.b.b.* (n $\geq$5 animals per group).
(TIFF)

**S1 Table. Primary antibodies used for immunolabelling tissue sections.**
(DOCX)

**S2 Table. Primers used for RT-PCR.**
(DOCX)

## Acknowledgments

We thank all the comments, suggestions, criticism and support from Prof Marina Bentivoglio and Prof Krister Kristensson. We also thank Ms Berit Olsson for the laboratory help and Ms Torun Söderberg and Ms Helen Braxenholm for the assistance in animal handling.

## Author Contributions

**Conceptualization:** Martin E. Rottenberg.

**Data curation:** Gabriela C. Olivera, Chiara Tesoriero.

**Formal analysis:** Gabriela C. Olivera, Leonie Vetter, Chiara Tesoriero, Federico Del Gallo, Gustav Hedberg, Martin E. Rottenberg.

**Funding acquisition:** Martin E. Rottenberg.

**Investigation:** Gabriela C. Olivera, Leonie Vetter, Chiara Tesoriero, Gustav Hedberg, Juan Basile.

**Methodology:** Gabriela C. Olivera, Leonie Vetter, Chiara Tesoriero, Federico Del Gallo, Gustav Hedberg, Juan Basile.

**Project administration:** Martin E. Rottenberg.

**Resources:** Martin E. Rottenberg.

**Software:** Juan Basile.

**Supervision:** Gabriela C. Olivera, Martin E. Rottenberg.

**Visualization:** Chiara Tesoriero, Federico Del Gallo, Juan Basile.

**Writing – original draft:** Martin E. Rottenberg.

**Writing – review & editing:** Gabriela C. Olivera, Leonie Vetter, Chiara Tesoriero, Federico Del Gallo, Martin E. Rottenberg.

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
