## [Decision Letter · Decision Letter 0]

26 May 2021

Dear Dr. Rottenberg,

Thank you very much for submitting your manuscript "Role of T cells during the cerebral infection with Trypanosoma brucei" for consideration at PLOS Neglected Tropical Diseases. As with all papers reviewed by the journal, your manuscript was reviewed by members of the editorial board and by several independent reviewers. In light of the reviews (below this email), we would like to invite the resubmission of a significantly-revised version that takes into account the reviewers' comments. Further experimentation will be welcomed but not required as long as the revised manuscripts can address the reviewers' comments.

We cannot make any decision about publication until we have seen the revised manuscript and your response to the reviewers' comments. Your revised manuscript is also likely to be sent to reviewers for further evaluation.

Sincerely,

Mary Ann McDowell

Associate Editor

Michael Boshart

Deputy Editor

Reviewer's Responses to Questions

**Key Review Criteria Required for Acceptance?**

**Methods**

-Are the objectives of the study clearly articulated with a clear testable hypothesis stated?

-Is the study design appropriate to address the stated objectives?

-Is the population clearly described and appropriate for the hypothesis being tested?

-Is the sample size sufficient to ensure adequate power to address the hypothesis being tested?

-Were correct statistical analysis used to support conclusions?

-Are there concerns about ethical or regulatory requirements being met?

Reviewer #1: The objectives of the study are clearly stated and the study design is appropriate. Sample size and statistics are adequate. The study is ethically approved as stated in the manuscript.

Reviewer #2: (No Response)

**Results**

-Does the analysis presented match the analysis plan?

-Are the results clearly and completely presented?

-Are the figures (Tables, Images) of sufficient quality for clarity?

Reviewer #1: The analysis matches the analysis plan. Figures are of high quality. Presentation of the results is clear.

Reviewer #2: (No Response)

**Conclusions**

-Are the conclusions supported by the data presented?

-Are the limitations of analysis clearly described?

-Do the authors discuss how these data can be helpful to advance our understanding of the topic under study?

-Is public health relevance addressed?

Reviewer #1: The conclusions are justified. Limitations of the study are not described. The relevance of the data is discussed. The experimental mouse study was not designed to address public health issues.

Reviewer #2: (No Response)

**Editorial and Data Presentation Modifications?**

Reviewer #1: See Summary and General Comments

Reviewer #2: (No Response)

**Summary and General Comments**

Reviewer #1: Olivera et al. studied the kinetics, functional status and anatomic location of CNS T cells in murine Trypanosoma brucei brucei infection. 

Based on their results the authors conclude that early upon systemic infection of mice with Tbb effector-memory T cells but at later stages resident memory T cells are present in the brain. T cells control Tbb but are also required for invasion of Tbb into the brain parenchyma. These data provide significant new information on the relation of T cells, brain invasion and spread of T.b.b., and clinical signs of sleeping disease in mice.

The authors used a broad spectrum of techniques including sophisticated mouse models to unravel the T cell-specific immune response in different compartments of the CNS during T.b.b. infection. Data are well presented in the figures, which have a high quality, and interpretations are plausible.

However, some important points remain, in particular to readability of the manuscript as detailed below.

Specific comments:

- For readers not familiar with the different anatomic structures of the CNS, i.e. the majority of PNTD readers, the manuscript is difficult to understand. E.g. in the Abstract the authors state that T cells are required for brain invasion of T.b.b. to the brain. Some few sentences later the authors report that parasite numbers were higher in the median eminence, a brain structure “with increased vascular permeability”, of rag-/- mice. WT mice. To avoid confusion for those readers not familiar with the different anatomic structures and the BBB it would be helpful, if the authors carefully revise the manuscript and provide better information on the different anatomic structures of the CNS and their immunological differences.

- In the same line, lines 209-217 need more explanations: what are tanycytes? Why do the authors stain for glucose transporters? It reads like the ME is not part of the brain parenchyma? Correct?

- Eventually a graphical illustration demonstrating T cells and T.b.b. in different CNS compartments over time would be helpful.

- Additionally, the Abstract does not cover several relevant aspects of the study including the experiments with CD4-/- mice. Please, refine the Abstract.

- Also lines 389-400: what is sleep/wake transitions and sleep bouts? Please provide better explanation.

- Line 404: point to – before or during? 

- CD4 T cell outnumber CD8 T cells by far in Tbb-infected WT mice. However, in Tbb-infected WT and CD4-/- mice, numbers of intracerebral CD8 T cells did not differ and IFNg levels were equal. In this context, the authors should discuss that the CD8 T cell compartment of CD4-/- contains MHC cl. II-restricted CD8 T cells and reduced numbers of MHC cl. I-restricted CD8 T cells and how this different MHC restriction of CD8 T cells of CD4-/- mice may affect the T cell response, pathogen control etc. Eg. See interpretation of data lines 256-257, 362-369.

Reviewer #2: In this work, the authors characterize the T cell compartment in the brain in comparison with deep cervical lymph nodes in mice infected with Trypanosoma brucei brucei (T.b.b) at several points of infection and with trypanocidal treatment. They further phenotypically characterize the T resident memory cells based on PD-1 expression. They confirm previous results showing that T cells are required for T.b.b. invasion of brain parenchyma but here they discard the involvement of CD4 T cells. They further show that there is no difference between the expression of factors related to endothelial cell function and inflammation between WT and Rag1 KO mice. Besides differences in IFNg production, Rag1 KO mice show less sleep alterations compared to WT mice.

This study shows some interesting findings such as the fact that CD4 T cells are not required for parasite or leukocyte invasion of the parenchyma and for IFNg production, and the role of lymphocytes in sleep alterations. However several findings are poorly characterized and connected between figures. Moreover, the discussion does not mention the role of other lymphocytes besides CD4 T cells in the brain, namely of CD8 T cells:

1) In Fig. 1 the authors show that trypanocidal treatment does not affect the number of T cells in the brain although they do not further characterize them based on PD1 expression. Furthermore, they do not show whether in treated mice there are alterations in the expression of inflammatory cytokines as done for Rag1 KO mice. Also, CCR7 expression used to distinguish central memory from effector memory cells is not shown here. In the Discussion, the authors mention low effector function of these cells but they have not performed any functional assay or have used markers of proliferation.

2) In Fig. 3 the authors demonstrate that CD4 T cells are not required for parasite or leukocyte invasion and for IFNg induction. This strongly suggests that CD8 T cells may play a fundamental role in the brain in infected mice. Did the authors try to treat WT mice with anti-CD8? The potential role of CD8 T cell in the infection should at least be addressed in the discussion.

3) In Fig. 4 the authors show that in Rag1 KO mice, besides differences in IFNg, the expression of other metabolism or inflammation-related genes is not affected. Is the expression of these genes dependent on IFNg? Why the authors are showing day 27 of infection? 

4) In Fig. 5 although the authors find differences in sleep patterns between WT and Rag1 KO mice, it would be interesting to know whether they are related to an altered function of the hypothalamus such as production of orexin/hypocretin. 

Other issues:

1) Legend of Figure 1 does not mention K. 

2) In line 158, “The levels of both TEM and TCM in the dcLN were increased after …” but the graph shows that an increase is only significant for TEM in case of CD4 T cells and TCM in case of CD8 T cells.

PLOS authors have the option to publish the peer review history of their article (what does this mean?). If published, this will include your full peer review and any attached files.

Reviewer #1: No

Reviewer #2: Yes: Teresa F Pais
---

## [Decision Letter · Decision Letter 1]

25 Aug 2021

Dear Dr. Rottenberg,

We are pleased to inform you that your manuscript 'Role of T cells during the cerebral infection with Trypanosoma brucei' has been provisionally accepted for publication in PLOS Neglected Tropical Diseases.

Best regards,

Mary Ann McDowell

Associate Editor

Michael Boshart

Deputy Editor

Reviewer's Responses to Questions

**Key Review Criteria Required for Acceptance?**

**Methods**

-Are the objectives of the study clearly articulated with a clear testable hypothesis stated?

-Is the study design appropriate to address the stated objectives?

-Is the population clearly described and appropriate for the hypothesis being tested?

-Is the sample size sufficient to ensure adequate power to address the hypothesis being tested?

-Were correct statistical analysis used to support conclusions?

-Are there concerns about ethical or regulatory requirements being met?

Reviewer #1: The authors have a clear scientific question and hypothesis: The role of T cell populations in brain invasion and intracerebral control of Trypanosoma brucei brucei. The study design is appropriate to address the stated objectives. The authors use several different mouse strains to test their hypothesis and, in general, all experimental groups have a sufficient number n to perform statistical analysis. The animal experiments have been approved by the respective authorities.

Reviewer #2: (No Response)

**Results**

-Does the analysis presented match the analysis plan?

-Are the results clearly and completely presented?

-Are the figures (Tables, Images) of sufficient quality for clarity?

Reviewer #1: The authors provide a series of carefully performed experiments which build logically on each other. The data are clearly and completely presented. The new Summary graphic nicely illustrates the key findings o the experiments. Overall, all figures have a high quality. Also the new experiments on the role of T cell subsets were adequately performed and described.

Reviewer #2: (No Response)

**Conclusions**

-Are the conclusions supported by the data presented?

-Are the limitations of analysis clearly described?

-Do the authors discuss how these data can be helpful to advance our understanding of the topic under study?

-Is public health relevance addressed?

Reviewer #1: The conclusions are clear and supported by the presented data. The clinical relevance of sleeping sickness is described and the authors carefully discuss how their experimental murine data improve our understanding of the human disease. The new experiments on T cell functions improved the manuscript and supported the conclusions.

Reviewer #2: (No Response)

**Editorial and Data Presentation Modifications?**

Reviewer #1: I have no concerns and critical points regarding the manuscript.

Reviewer #2: (No Response)

**Summary and General Comments**

Reviewer #1: The authors carefully revised the manuscript and performed an additional series of experiments, which greatly improved the manuscript and addresses all reviewers questions. The inclusion of a summarizing graphic makes the data easy to understand and improved the manuscript. Since all reviewer's questions have been sufficiently addressed. I have no more concerns.

Reviewer #2: (No Response)

PLOS authors have the option to publish the peer review history of their article (what does this mean?). If published, this will include your full peer review and any attached files.

Reviewer #1: No

Reviewer #2: **Yes: **Teresa F. Pais

---

## [Editor Report · Acceptance letter]

13 Sep 2021

Dear Dr. Rottenberg,

We are delighted to inform you that your manuscript, "Role of T cells during the cerebral infection with Trypanosoma brucei," has been formally accepted for publication in PLOS Neglected Tropical Diseases.

Best regards,

Shaden Kamhawi

co-Editor-in-Chief

Paul Brindley

co-Editor-in-Chief
